# Implementation of an Intelligent Trap for Effective Monitoring and Control of the *Aedes aegypti* Mosquito

**DOI:** 10.3390/s24216932

**Published:** 2024-10-29

**Authors:** Danilo Oliveira, Samuel Mafra

**Affiliations:** Instituto Nacional de Telecomunições (INATEL), Santa Rita Sapucai 37536-001, Minas Gerais, Brazil; danilomachado@mtel.inatel.br

**Keywords:** *Aedes aegypti*, IoT, smart trap

## Abstract

*Aedes aegypti* is a mosquito species known for its role in transmitting dengue fever, a viral disease prevalent in tropical and subtropical regions. Recognizable by its white markings and preference for urban habitats, this mosquito breeds in standing water near human dwellings. A promising approach to combat the proliferation of mosquitoes is the use of smart traps, equipped with advanced technologies to attract, capture, and monitor them. The most significant results include 97% accuracy in detecting *Aedes aegypti*, 100% accuracy in identifying bees, and 90.1% accuracy in classifying butterflies in the laboratory. Field trials successfully validated and identified areas for continued improvement. The integration of technologies such as Internet of Things (IoT), cloud computing, big data, and artificial intelligence has the potential to revolutionize pest control, significantly improving mosquito monitoring and control. The application of machine learning (ML) algorithms and computer vision for the identification and classification of *Aedes aegypti* is a crucial part of this process. This article proposes the development of a smart trap for selective control of winged insects, combining IoT devices, high-resolution cameras, and advanced ML algorithms for insect detection and classification. The intelligent system features the YOLOv7 algorithm (You Only Look Once v7) that is capable of detecting and counting insects in real time, combined with LoRa/LoRaWan connectivity and IoT system intelligence. This adaptive approach is effective in combating *Aedes aegypti* mosquitoes in real time.

## 1. Introduction

The *Aedes aegypti* mosquito, which transmits diseases such as dengue, Zika, and chikungunya, is a growing threat to public health, especially in tropical and subtropical regions. Originally from Africa, this mosquito has adapted well to urban environments, where it proliferates in areas with stagnant water. Factors such as the rapid growth of cities, inadequate waste disposal, and climate change have contributed to the spread of this vector in densely populated areas [1,2].

Data from the Ministry of Health show a worrying increase in deaths and cases of dengue, which makes it urgent to seek innovative and effective solutions to control this mosquito [3]. Traditional epidemiological surveillance actions, which include inspections of homes and public spaces, are susceptible to human error. In addition, the frequent use of pesticides has negative impacts on both human health and the environment, reinforcing the need for more sustainable and accurate alternatives [4,5].

In recent years, the use of advanced technologies such as the Internet of Things (IoT), cloud computing, big data, and artificial intelligence has emerged as a promising solution to improve the monitoring and control of the *Aedes aegypti* mosquito. These technologies offer tools that increase the efficiency of real-time data collection and analysis, allowing for more precise and efficient control of *Aedes aegypti* populations [3,6].

This work proposes the development of a smart trap, equipped with high-resolution cameras and advanced machine learning algorithms, for the detection and selective capture of winged insects, especially *Aedes aegypti*. The trap uses the YOLOv7 algorithm, which allows real-time detection, and combines LoRa/LoRaWan connectivity with IoT intelligence for continuous monitoring. This adaptive system offers an innovative and effective solution to combat the mosquito, allowing control actions to be faster and more precise [2,7,8,9].

The structure of this paper is as follows: Section 2 discusses related work in the development of smart traps and the application of advanced technologies for mosquito control. Section 3 outlines the proposed system architecture, including the integration of IoT devices, ML algorithms, and communication protocols. Section 4 presents the experimental setup and the results obtained, highlighting the effectiveness of the YOLOv7 algorithm. Finally, Section 5 concludes the paper with a discussion of the findings and suggestions for future works.

## 2. Related Work

### 2.1. Biology and Behaviour of *Aedes aegypti*

Understanding the biology of the *Aedes aegypti* mosquito is essential to formulate effective control strategies. In this section, elements related to the morphology, physiology, life cycle, and behavior of this vector will be discussed. *Aedes aegypti*, belonging to the Culicidae family, is known for its propensity to spread viral diseases such as dengue, Zika, and chikungunya. Anatomically, it is characterized by its sensory antennas, mouthparts adapted for skin piercing, and wings that allow swift flight.

*Aedes aegypti*, commonly known as the dengue mosquito, has a fascinating holometabolic life cycle, encompassing terrestrial and aquatic phases. The study of [10] detailed the complex life journey of *Aedes aegypti*, from egg to adult, along with its fundamental role in viral transmission and the onset of disease symptoms, as illustrated in Figure 1. The cycle begins with the pupal stage (Figure 1a) , passes through the larvae (Figure 1d), on to the eggs (Figure 1c), and then to the adult stage of the mosquito (Figure 1b). When female *Aedes aegypti* mosquitoes find suitable conditions, they meticulously lay their eggs on the inside walls of containers filled with clean water, strategically above the water line. These resistant eggs can withstand long periods of desiccation, waiting for the moment when humidity triggers hatching, thus releasing the first instar larvae. The development of the larva, a crucial phase that lasts between 7 and 14 days, is intrinsically linked to environmental factors such as food availability and temperature. From the first instar larva, there is rapid progression to the third instar, while the fourth instar larva undergoes a prolonged period of growth, significantly increasing in size and weight before transitioning to the pupal stage [11]. Transmission of the virus occurs when an infected human (indicated in the image) is bitten by the mosquito (Figure 1e), causing the mosquito to become a carrier of viruses such as dengue, chikungunya, or Zika. These viruses are shown in Figure 1f, Figure 1g and Figure 1h, respectively. After another bite on a healthy human, the virus is transmitted.

The pupal stage serves as a critical metamorphosis phase that bridges the larval and adult stages, usually lasting 2 to 3 days without feeding until the adult mosquito emerges. After emergence, the adult mosquito, characterized by its distinctive appearance and limited aerial mobility (Figure 1b), enters the reproductive phase, feeding mainly on plant nectar during its early stages. After copulation, gravid females undergo a change in food preference, seeking blood meals as a protein-rich resource for egg production (Figure 1e) [12].

However, it is during these blood meals that the *Aedes aegypti* mosquito plays a key role in viral transmission. Ingestion of blood from an infected human being causes the mosquito to become a carrier of viruses, which penetrate the cells of the midgut and subsequently spread to secondary organs and salivary glands [13,14]. After subsequent blood feeding from a healthy individual, these viruses are injected into the bloodstream, initiating a cascade of events in the human host.

Once in the human body, these viruses cause oxidative stress via NADPH oxidase, thus triggering a robust inflammatory cytokine response [11,15,16]. The resulting release of pro-inflammatory cytokines is implicated in the manifestation of common symptoms such as fever, weakness, headache, and pain (Figure 1f). In rare cases, the heightened inflammatory response can lead to increased capillary permeability and hemorrhage (Figure 1g), culminating in severe dengue [15]. Furthermore, the exact mechanism underlying the association between Zica infection and microcephaly remains elusive. However, it is postulated to involve an intricate immune response in the newborn brain, inducing inflammatory cascades, neuronal apoptosis, and cell death, potentially triggered by both the virus and the host immune system [11,17] (Figure 1).

To attract the insect of interest, a pheromone, octenol, is placed inside the trap. As illustrated in Figure 1, an internal camera in the entry tube detects the insects. After detection, insect selection is performed. For example, when identifying an *Aedes aegypti* mosquito, the cooler is activated, as shown in Figure 1b, creating a wind tunnel that directs the mosquito into a container containing a viscous liquid, where it will be trapped. In the case of a bee or another pollinating insect, the insect is directed out of the trap, as shown in Figure 1a, thus preserving the pollinator species.

### 2.2. Smart Traps and Pest Control

The growing demand for innovative pest control methods has led to the integration of artificial intelligence (AI) and Internet of Things (IoT) technologies. Smart traps have emerged as a sustainable solution to monitor and control pests by leveraging real-time data for effective pest management. Various traditional techniques for monitoring insects include visual inspection and traps with pheromones or food baits, commonly used in agriculture and forestry. These methods aim to monitor and predict insect population density and damage potential [18]. Since the early 20th century, researchers have focused on detecting insects through wingbeat frequency analysis, which correlates wingbeat frequency with insect size [19,20]. Optical methods, such as infrared beam interruption and light scattering, have also been used to track insects entering traps [21].

Several recent studies have explored different approaches for mosquito detection and capture, utilizing various technologies ranging from traps equipped with ultraviolet lights to digital monitoring systems and robots with artificial intelligence for species classification. A notable example is the study that used the Dragonfly robot, designed to capture and map mosquito populations, demonstrating high accuracy in species classification through the use of deep neural networks [22]. Another study compared four types of traps, highlighting the superior efficiency of the BLT trap in both mosquito capture and species diversity collected [23]. However, the present work significantly advances the state of the art by integrating an innovative IoT system with deep learning algorithms.

Unlike existing methods, which mainly focus on capture or species classification, our system is capable of capturing and analyzing in real-time the movement patterns and population density of mosquitoes. This capability not only provides immediate data for public health interventions but also offers a more detailed and contextualized analysis of mosquito behavior, enabling more effective control of diseases transmitted by these vectors. This approach represents a significant advancement over manual methods and other automated solutions mentioned in the literature, establishing itself as a powerful tool for mosquito monitoring and control in urban and rural areas [24].

In recent years, advances in acoustic and machine learning techniques have enhanced insect identification based on wingbeat frequencies. For example, experiments have focused on classifying mosquitoes like *Aedes aegypti* by analyzing their flight tones, with studies demonstrating wingbeat frequencies ranging between 480 Hz and 511 Hz [25]. Such information has been critical in developing automated traps, like the McPhail electronic trap, which detects insect wingbeat frequencies through sensors [26].

Recent work in deep learning and computer vision has also contributed significantly to mosquito detection and classification. For example, convolutional neural networks (CNNs) and YOLOv7 algorithms have been successfully applied for real-time identification of *Aedes aegypti*, aiding in disease vector control [27,28]. Unmanned aerial vehicles (drones) have further enhanced monitoring by identifying mosquito breeding sites using image processing and classification [28,29]. These AI-based technologies are promising tools in modern vector control strategies.

Despite advancements, challenges remain, particularly in detecting mosquitoes at different life stages and environments. Therefore, automated systems complement, rather than replace, traditional vector control methods. The research in [30] investigates the detection of floating and underwater naval mines using images captured by drones, submarines, ships, and boats. Due to the scarcity of real images, synthetic images were generated by overlaying mines on water backgrounds. Two datasets were created: one for floating mines and another for underwater mines. Three deep learning models (YOLOv5, SSD, and EfficientDet) were trained and compared, with YOLOv5 and SSD applied to floating mines, and YOLOv5 and EfficientDet to underwater mines.

In the context of communication and surveillance, Fort, Peruzzi, and Pozzebon [31] developed a near real-time video surveillance unit using LoRaWAN for real-time image transmission, aimed at remote monitoring in security and protection domains. This study demonstrates that the integration of low-power communication technologies with computer vision systems is essential for the development of smart traps, which require efficient and reliable connectivity.

Dhou et al. [32] also contributed to this field by exploring the use of mobile sensors based on machine learning and IoT to assist visually impaired people. This work exemplifies the practical impact of intelligent sensors and machine learning algorithms in improving quality of life, highlighting the potential of these technologies in a wide range of applications. Pronello and Ruiz [33] conducted a comparative study on the performance of automated video-based passenger counting systems under real-world conditions. The performance and accuracy analysis of these systems in operational environments is highly relevant to our research as we seek to evaluate the effectiveness of computer vision systems in challenging contexts, similar to those found in mosquito detection.

## 3. Methodology

This section proposes a technological solution for pest control aimed at detecting the *Aedes aegypti* mosquito. The solution has several blocks, such as hardware, computer vision models, and user interface, which will be presented in the following subsections.

### 3.1. Architecture

The architecture proposed, as shown in Figure 2, is an intricate assembly of components designed for the real-time surveillance and identification of *Aedes aegypti*. The Raspberry Pi 4 forms a crucial part of this architecture, serving as the main executor of the YOLOv7 real-time detection algorithm. The camera, which interfaces with the Raspberry Pi 4, captures images of flying insects. These images are then processed by the algorithm to identify and categorize the insects present. The capability of performing this detection in real time is a significant feature of the smart trap proposal, facilitating the location and classification of flying insects, and thus enhancing the effectiveness and efficiency of the detection system.

The smart trap has a global positioning system (GPS) module to enhance the efficiency and accuracy of locating these traps. The use of GPS in selective traps for winged insects offers a series of significant advantages. Firstly, it allows real-time monitoring of the exact location of the traps, which is crucial for analysis and quick decision making. In addition, it facilitates the collection of geospatial data that can be used to map insect distribution patterns and optimize control strategies.

The ESP32 LoRa is interfaced with the Raspberry Pi board, facilitating communication between the two devices. This connection enables data and information exchange, integrating additional functionality into the system. The ESP32 LoRa can function as a transmitter of the collected data or serve as an auxiliary temperature and humidity sensor, providing supplementary information to the system. This integration of components improves the capabilities of the system, paving the way for the development of more comprehensive and efficient solutions. The Table 1 shows the parameters of the architecture.

Upon detection and processing of the data, transmission is carried out using LoRa technology, which offers long-range communication while minimizing power consumption. The LoRaWAN protocol is used for this transmission, encapsulating the gathered data. The gateway functions as the receiver of the data transmitted from the device, acting as an intermediary between the device and the LoRaWAN network server. It receives and decodes the data packets before forwarding them to the network server.

An application was created in the user interface to show information about smart trap captures. The Node-Red was used to create this interface. The Node-Red was installed on a Raspberry Pi board, which had sufficient computing capacity to execute locally and in parallel all the functions and capabilities present on the platform. In Node-RED, it is possible to create workflows (flows) that receive GPS data, process them, and present them in a useful and accessible manner. For example, you can set up a flow to visualize the locations of traps on an interactive map, allowing users to easily monitor their distribution and movement.

### 3.2. Hardware

#### Prototyping the Smart Trap

It is essential to highlight the importance of the detection and selective capture of female *Aedes aegypti* mosquitoes. With this in mind, a dodecahedron-shaped trap was developed, with an approximate volume of 300 cm^3^, as illustrated in Figure 3. The structure has five internal tubes with a diameter of 50 mm. Four of these tubes function as forced ventilation ducts.

When detection of bees, butterflies, or other pollinating insects that play a significant role in nature and food production occurs, fans (c) and (d) located at the back of the trap are activated, creating an airflow that expels the detected insect from the structure. Upon detection of an *Aedes aegypti* mosquito, front fans (a) and (b) are activated, creating an airflow that directs the mosquito towards a reservoir containing pheromones, resulting in its capture.

The inlet and outlet tubes are equipped with a hybrid vision camera capable of capturing real-time images of *Aedes aegypti*. This camera combines night vision and day vision capabilities, allowing image capture in a wide range of lighting conditions. Night vision enables image capture in low-light environments, whereas day vision is ideal for daytime recording. In Figure 4, various perspectives of the trap are presented. In Figure 4a, the ventilation vents and the insect entry and exit openings are highlighted. In Figure 4b, the assembly and its connections can be examined. Figure 4c illustrates the access door, used both for insect removal and for the replacement of adhesive tape or pheromone liquid.

Figure 5 shows the operation of the trap in two stages, highlighting its main electronic and communication components.

(a) Initial State: The trap detects the presence of bees by using a camera attached to a Raspberry Pi 4 board, with the YOLOv7 algorithm running, which is responsible for identifying the bees, and the CPU activates the fans, generating an airflow that directs the insect towards the trap entrance. The camera captures images for monitoring, and the LoRa antenna transmits data about the capture and the trap’s status to the base station. The rechargeable batteries ensure the continuous operation of all components.

(b) Activated State: Upon detecting an *Aedes aegypti* mosquito, the CPU again activates the fans, directing the mosquito into the capture container with pheromone, which prevents its escape. The camera records the process, and the LoRa antenna sends real-time information, including data about the capture and the battery level.

The dimensions of the trap are indicated in Figure 5, with a width of 120 mm, providing a clear reference for the size and scale of the device.

### 3.3. Computer Vision Model

#### 3.3.1. Data for Insect Detection and Classification

For the training process, a dataset was compiled containing images of various insect species, including *Aedes aegypti*, bees, and butterflies. The proposed application was designed to monitor and capture the *Aedes aegypti* mosquito. However, to reduce the likelihood of false alarms, the dataset was expanded to include three categories. Bees and butterflies were selected as additional categories due to their frequent presence in agricultural environments [25,28,34]. Creating this data set was crucial to developing insect detection and classification models essential for pest control and biodiversity preservation.

The images were acquired by merging datasets available in the *Kaggle* community, an online data science platform. The result was a comprehensive dataset consisting of 7673 images. Among them, 3371 belonged to the *Aedes aegypti* class [35], 3637 to the bee class, and 665 to butterflies. Examples of the image dataset are illustrated in Figure 6. The imbalance in the dataset was managed by recognizing that there were fewer images of butterflies compared to the *Aedes aegypti* and bee classes. To reduce this problem, preprocessing techniques and algorithm adjustments were applied to ensure the accuracy and generalization of the model. The dataset was split into 70% for training, 20% for validation, and 10% for testing, ensuring adequate representation of all classes.

An important step in data preparation was image normalization, as the original images were of different sizes. Normalization was performed to standardize all photos to a resolution of 640 × 640 pixels. This ensured that all images were of the same size and facilitated further processing.

Additionally, to create the annotated dataset, an image data annotation program was used [36], allowing for the manual labeling of each image, assigning classification information for each insect (bee, butterfly and *Aedes aegypti*. This classification information was then stored in text files, ensuring the integrity and traceability of the data. The creation of this annotated dataset is of great importance as it provides the necessary information to train and evaluate insect detection and classification models.

#### 3.3.2. YOLOv7

The innovation and originality in this study lies in the combination of advanced technologies such as the Internet of Things (IoT), cloud computing, big data and artificial intelligence. The system uses these resources to create a smart trap capable of monitoring in real time the presence of the *Aedes aegypti* mosquito, a vector of diseases such as dengue, Zika, and chikungunya. One of the main innovations is the use of the YOLOv7 (You Only Look Once) computer vision algorithm, which allows the identification and counting of insects in real time with high precision, using high-resolution cameras and sensors.

In addition, the system adopts a sustainable approach, selectively capturing the *Aedes aegypti* mosquito without affecting other beneficial insects, such as bees and butterflies, through a ventilated ventilation system. Communication between devices is carried out through LoRa/LoRaWAN technology, which offers long-range communication with low power consumption, allowing the use of the trap in hard-to-reach urban and rural areas. These elements combined make the system a highly innovative, efficient, and sustainable solution for mosquito control, representing a significant advance in strategies to combat vector-borne diseases.

The algorithm classifies detected insects into appropriate categories. Figure 7 demonstrates images from onboard camera. If an *Aedes aegypti* mosquito is detected, the fans direct the mosquito into the capture reservoir. If a non-target insect (such as a bee) is detected, the fans redirect the insect out of the trap.

The flowchart shown in Figure 8 describes an insect detection and treatment process, starting with the insect entering the system, a structure equipped with sensors and cameras. The first step is to detect the presence of the insect using a sensor, possibly aided by the YOLOv7 algorithm, chosen for its real-time accuracy and speed, operating between 5 and 160 frames per second with 56.8% accuracy. After this detection, the system checks whether the insect is a bee. If it is identified as such, the insect is redirected out of the environment, due to the ecological importance of bees. If the insect is not a bee, the system proceeds to check whether it is an *Aedes aegypti* mosquito, known to transmit diseases such as dengue and Zika. The identification is performed with the help of YOLOv7 and its E-ELAN architecture, which improves the system’s learning capacity, ensuring accurate detection, even in variable conditions. If the presence of *Aedes aegypti* is confirmed, the system performs a specific capture of this mosquito to prevent the spread of diseases. In the event that the insect is neither a bee nor an *Aedes aegypti* mosquito, it is also redirected outside. After each redirection or capture operation, the system continues to monitor the environment, providing feedback and preparing to treat new insects that may enter, ending the cycle [37,38].

### 3.4. User Interface

The smart trap was deployed to detect and identify these insects, providing data for decision making regarding control and monitoring. For this purpose, the solution implemented was The Things Network (TTN), which includes the ChirpStack LoRaWAN network server stack, as shown in Figure 9.

Through the TTN interface, it is possible to visualize the location of devices on a map. This allows users to track the location of devices and monitor the flow of data between gateways and the network. The information about the captures is displayed in a Node-RED interface. The Node-Red enables easy integration of IoT devices and services, making it an ideal choice for connecting GPS to selective traps. Additionally, Node-RED can be programmed to send alerts or notifications when certain criteria are met, such as the detection of a high concentration of insects in a specific area.

The Node-Red tool was used as the development environment for the back-end of the solution, where all the integration between different systems was carried out, in addition to implementing all data processing, from reception from the smart trap to the presentation of the results to the end user. Node-RED features a specific set of nodes to connect to TTN and receive data sent by LoRaWAN devices. These nodes allow configuring access credentials to TTN, such as the App EUI and App Key, and defining topics of interest to receive the data. Figure 10 depicts the established flow that aims to present essential information on the user interface (UI). This information includes the recorded temperature and humidity, the total number of detected insects, their respective classification, as well as the visualization of the location of the trap on a map.

In the dashboard presented in Figure 11, the user can visualize and monitor the real-time status of the trap, which is crucial to making decisions, especially in environments sensitive to climatic variations. In the bottom-right corner of the image, the total number of detected insects is displayed. This metric provides valuable information about the presence and magnitude of pests or insects of interest, contributing to a more in-depth analysis of the detection scenario.

## 4. Experimental Results

A growing threat to public health is posed by the *Aedes aegypti* mosquito, a vector of diseases such as dengue, Zika and chikungunya. The proliferation of this mosquito is driven by factors such as urbanization, improper waste disposal, and climate change, highlighting the need for innovative and effective solutions to control this vector. In the present study, the development of a smart trap that combined IoT devices, high-resolution cameras, and advanced machine learning algorithms for insect detection and classification included the use of a Raspberry Pi 4 to run the YOLOv7 detection algorithm in real time, with communication facilitated by LoRa/LoRaWAN technology, which offered long-range communication with low power consumption. Table 2 displays field tests carried out in Santa Rita do Sapucaí, MG, Brazil, which showed remarkable accuracy in detecting *Aedes aegypti* (97%), identifying bees (100%) and classifying butterflies, (90.1%).

The trap was strategically positioned in public squares and near streams in the Santa Rita do Sapucaí MG region, as illustrated in the images in Figure 12 and in the location indicated on the map in Figure 13. The trap was installed and monitored in the late afternoon, when there was a higher incidence of mosquitoes. During the testing period, the number of dengue cases was significantly higher than in any other period of the year, even surpassing the peak recorded in 2023. During this period, a solution of sugar water and octenol was placed in the trap to attract the mosquitoes. During the period, 20 *Aedes aegypti* mosquitoes were captured.

In the map shown in Figure 13, the locations where traps were installed to monitor potential breeding sites for the *Aedes aegypti* mosquito are presented. We can observe on the map that various locations have been selected. This contributes significantly to the efforts of health agents and epidemiology personnel in combating mosquito breeding sites.

This visual representation of traps on the map facilitates the identification of areas where the presence of the mosquito may be more prevalent. With this information at hand, health professionals can plan and direct their control and prevention efforts more effectively.

During field training with the smart trap to capture *Aedes aegypti* mosquitoes, we encountered some challenges related to the heating of the Raspberry Pi 4 board, resulting in operating system failures. To overcome these issues, we implemented measures such as using specific static and active heat sinks for the Raspberry Pi 4. These heat sinks helped reduce the temperature of the board, ensuring a more stable and reliable system operation.

Additionally, we replaced the original Raspberry Pi 4 camera module OV5647 with a Raspberry Pi OV5647 5MP 1080p camera with night vision. This change proved to be crucial for improving the quality of captured images and videos, especially in low-light or no-light environments. The night vision camera featured powerful infrared LEDs designed to illuminate the environment in the absence of visible light, thus ensuring better detection and monitoring of mosquitoes, even during the night.

Continuing with field tests, to evaluate LoRa communication, we initially used a 2.5 dBi antenna. However, we observed that the transmission rate was below expected levels, suggesting inefficiency in signal propagation. Furthermore, we noted a significant data packet loss rate, indicating a failure in communication integrity. Faced with these setbacks, we decided to replace the antenna with a 6 dBi antenna to optimize system efficiency and reliability, as demonstrated in Table 3.

Point A: This point is located in an area with moderate obstacles, such as some walls and furniture between the communication device (ESP32 LoRa) and the gateway. The distance between the device and the gateway is approximately 20 m in a straight line.Point B: This point is located in an area with significant obstacles, such as thick walls and interfering electronic equipment. The distance between the device and the gateway is approximately 30 m with direct obstacles in the path.Point C: This point is located in an area with minimal obstacles, with a clear line of sight between the device and the gateway. The distance between the device and the gateway is approximately 40 m in a straight line, without significant obstacles in the path.

The implementation of the model for the detection and classification of winged insects, using the YOLOv7 algorithm, proved to be a robust choice, recognizing its efficiency and precision in image analysis. In-depth analysis of the results, considering performance metrics, provided substantial insight into the model’s capabilities and strengths in different environments.

The synergistic integration of the solution with LoRa/LoRaWAN technology, particularly on The Things Network (TTN) platform and through the Node-RED application, added an additional layer of sophistication by enabling real-time remote monitoring of the smart trap. The ability to instantly visualize data and map device locations offered a dynamic perspective for pattern and behavior analysis.

However, the observation of results revealed certain limitations. Class identification, especially between butterflies and the background, posed challenges, highlighting the need for continuous refinement and optimization of the model. Additionally, the model’s sensitivity to environmental variations points to a critical area requiring meticulous attention.

## 5. Conclusions

The proposed approach has some limitations. Firstly, the model may face challenges in dynamic environments where climatic variations impact detection effectiveness. Additionally, there is a need to increase the diversity and volume of the dataset, which would allow for greater model generalization. Future research should focus on refining the model to improve the discrimination between insect classes and explore more advanced machine learning techniques, such as complex neural networks or transfer learning methods.

The results were quantitatively obtained during field tests carried out in Santa Rita do Sapucaí, Minas Gerais. In the laboratory, the detection model demonstrated remarkable accuracy, reaching 97% for *Aedes aegypti*, 100% for bees, and 90.1% for butterflies. During field tests, traps were strategically positioned in public squares and near streams, with their location monitored in real time using The Things Network (TTN) platform and the Node-RED application. Communication via LoRa was evaluated with two different antennas. The 2.5 dBi antenna presented an RSSI of −85 dBm and a data rate of 100 kbps at Point A, −90 dBm and 80 kbps at Point B, and −95 dBm and 60 kbps at Point C. The 6 dBi antenna showed better performance, with an RSSI of −80 dBm and a data rate of 120 kbps at Point A, −75 dBm and 150 kbps at Point B, and −70 dBm and 180 kbps at Point C. These results highlight the effectiveness of the model for detecting and classifying insects, especially the *Aedes aegypti* mosquito, and the importance of efficient communication via LoRa for real-time monitoring.

The study provided a foundation for advancements in the interaction of artificial intelligence, computer vision, and winged insect monitoring. Some directions for future work include:Model Refinement: Enhance the model’s ability to discriminate between classes, aiming for more precise identification, particularly when there is visual overlap or significant lighting variations;Expansion of Dataset: Increase the diversity and volume of the dataset, covering a wider range of environmental conditions and variations in insect characteristics, for greater model generalization;Adaptation to Dynamic Environments: Investigate strategies to increase the model’s robustness in dynamic environments, where variable weather conditions may impact detection effectiveness;Exploration of Advanced Techniques: Explore the application of advanced machine learning techniques, such as using more complex neural networks or transfer learning methods, to improve the model’s efficiency;Integration of Other Data Sources: Explore the integration of data from different sources, such as additional sensors or weather information, to enrich the analysis of insect behavioral patterns.

## Figures and Tables

**Figure 1 sensors-24-06932-f001:**
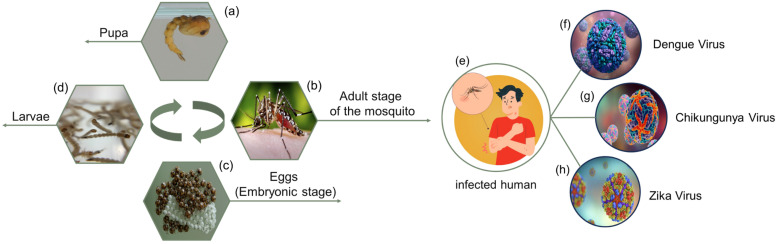
The life cycle of the *Aedes aegypti* mosquito and the transmission of viral diseases.

**Figure 2 sensors-24-06932-f002:**
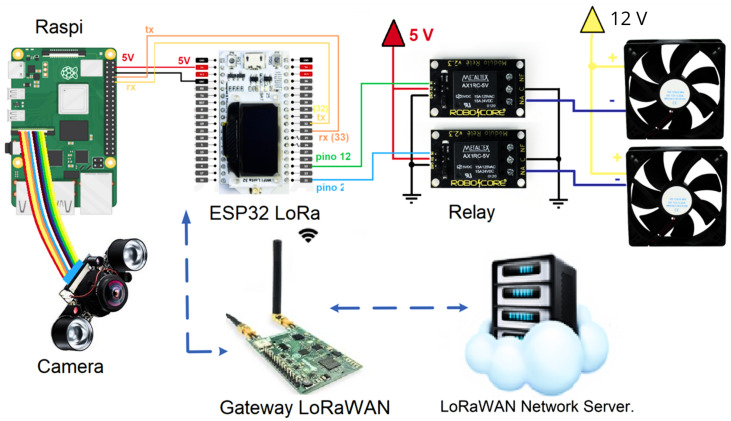
System architecture of the proposed IoT system.

**Figure 3 sensors-24-06932-f003:**
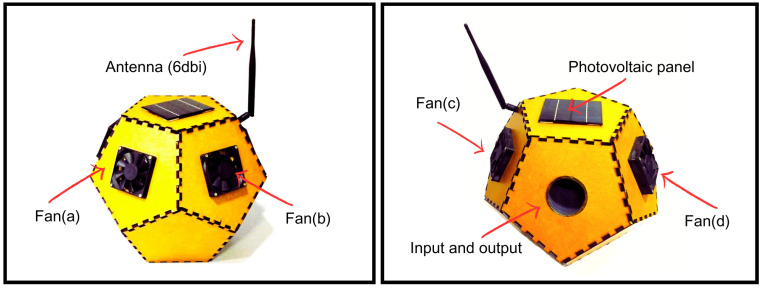
Prototype of smart trap.

**Figure 4 sensors-24-06932-f004:**
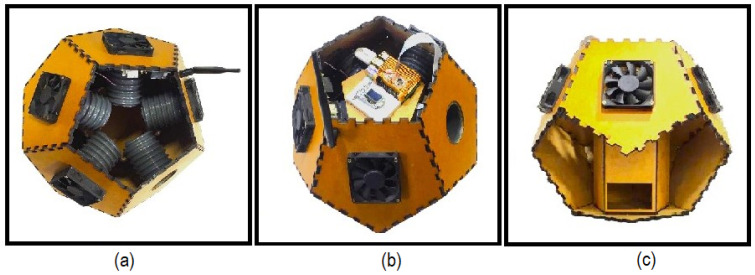
Illustration of perspectives of the smart trap, highlighting its components: (**a**) Ventilation ducts, as well as the entry and exit openings for insects; (**b**) Hardware and its connections; (**c**) Access door, used for both insect removal and replacement of adhesive tape or pheromone liquid.

**Figure 5 sensors-24-06932-f005:**
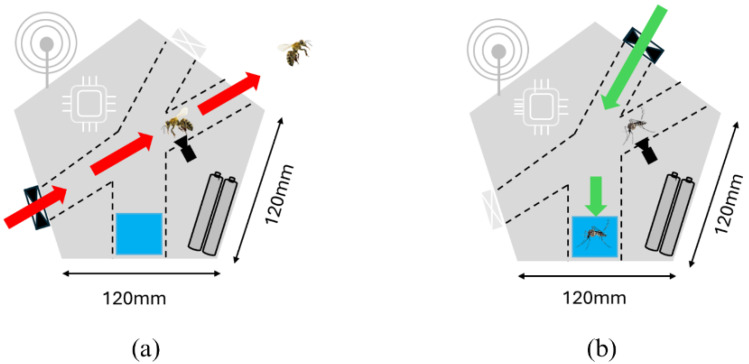
Internal view of the trap. Internal View of the trap: (**a**) represents the the detection of a bee and (**b**) the detection of an *Aedes aegypti* mosquito.

**Figure 6 sensors-24-06932-f006:**
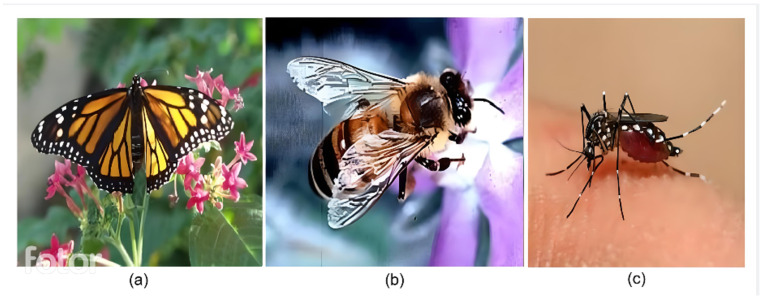
Images from data set of (**a**) butterfly; (**b**) bee; and (**c**) *Aedes aegypti*.

**Figure 7 sensors-24-06932-f007:**
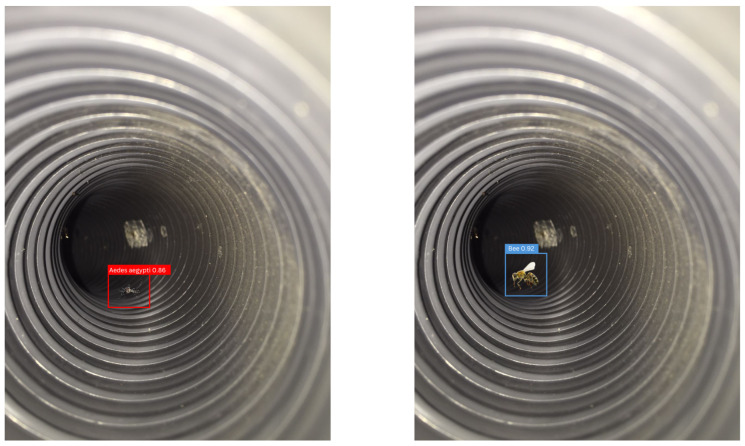
Images from onboard camera.

**Figure 8 sensors-24-06932-f008:**
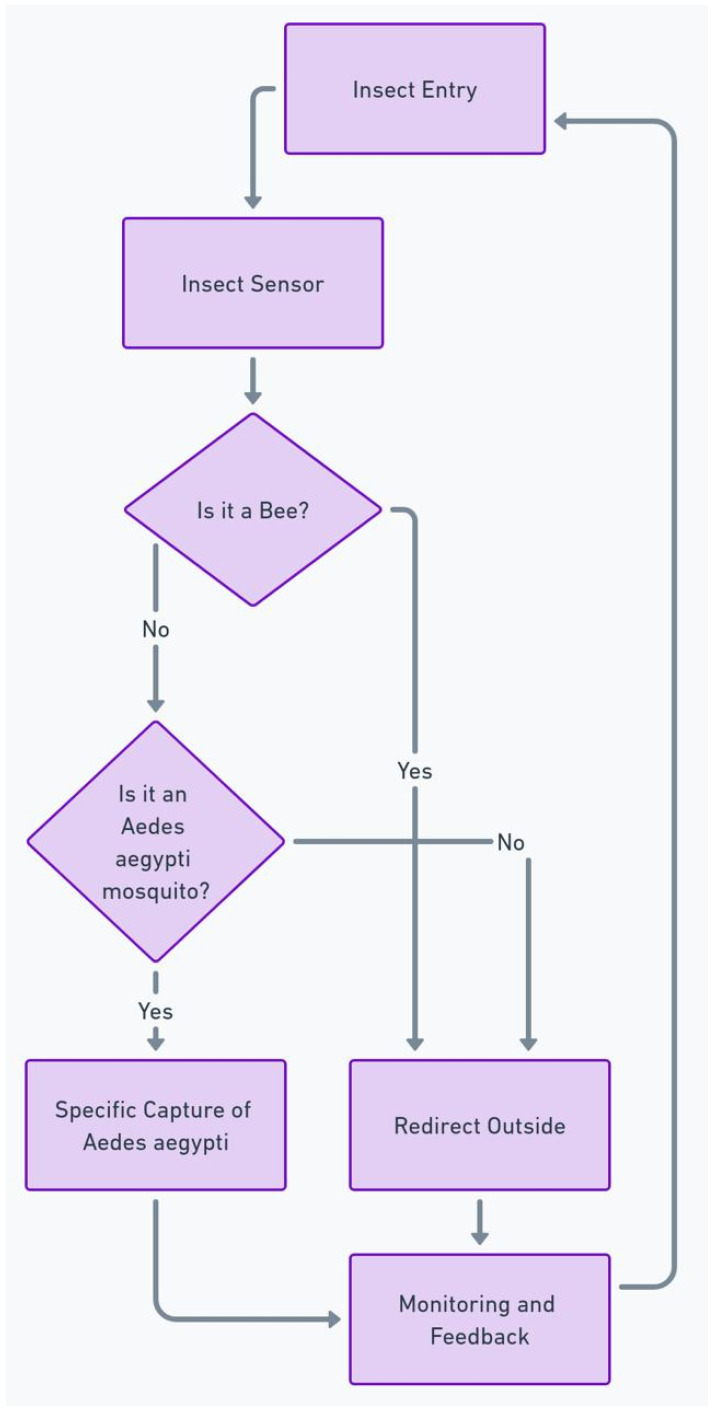
Flowchart.

**Figure 9 sensors-24-06932-f009:**
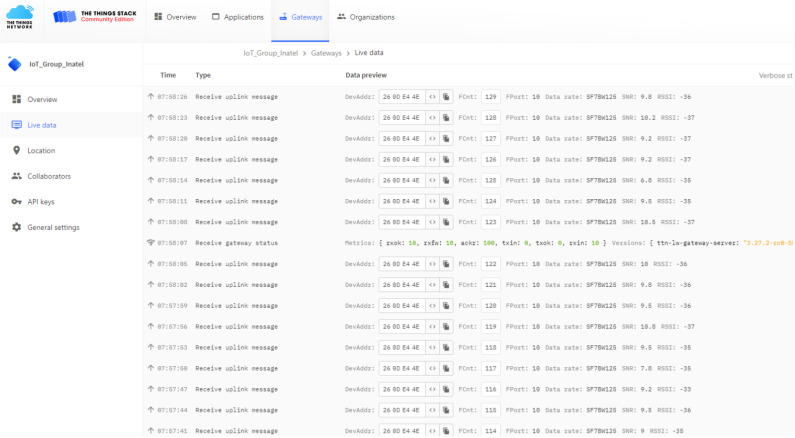
The Things Network (TTN).

**Figure 10 sensors-24-06932-f010:**
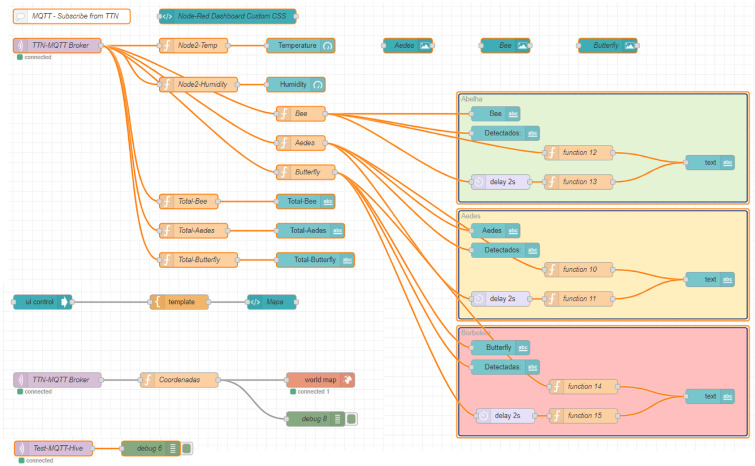
Node-RED flow.

**Figure 11 sensors-24-06932-f011:**
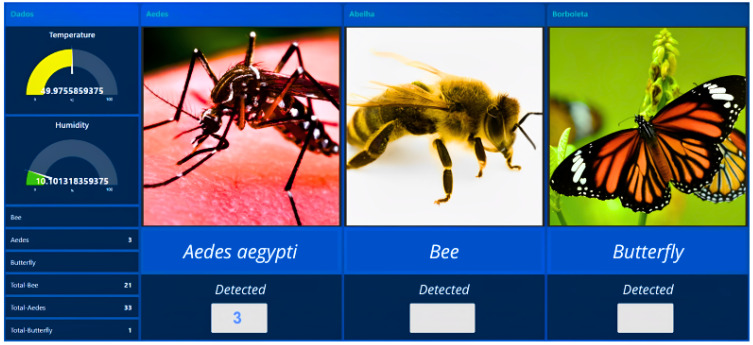
Temperature and humidity monitoring dashboard and real-time detection of winged insects.

**Figure 12 sensors-24-06932-f012:**
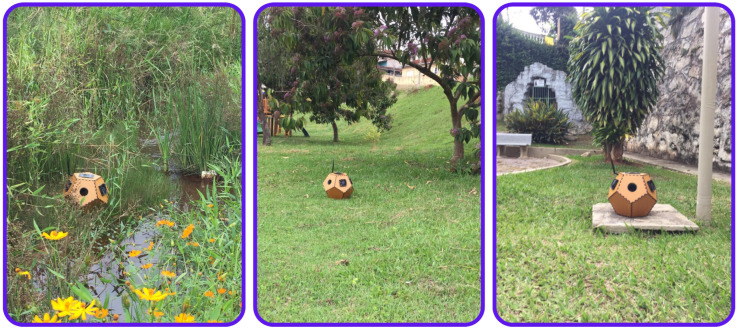
Traps in squares and streams in Santa Rita do Sapucaí-MG.

**Figure 13 sensors-24-06932-f013:**
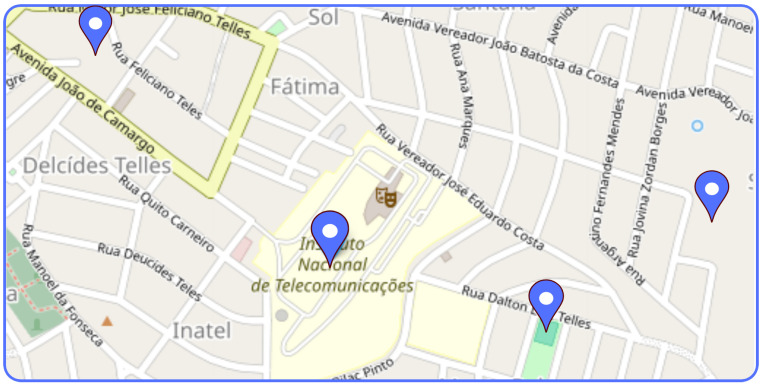
Trap location map.

**Table 1 sensors-24-06932-t001:** LoRa transmission parameters.

Parameter	Value
Transmitter Output Power	14 dBm
Spreading Factor (SF)	14 dB
Coding Rate (CR)	4/5
Bandwidth (BW)	125 kHz
Payload Length	320 bytes

**Table 2 sensors-24-06932-t002:** Results as a confusion matrix.

Real Class	*Aedes aegypti*	Bee	Butterfly	Total
* **Aedes aegypti** *	97	0	3	100
**Bee**	0	100	0	100
**Butterfly**	9.9	0	90.1	100
**Total**	106.9	100	93.1	300

**Table 3 sensors-24-06932-t003:** Field test results with 2.5 dBi and 6 dBi antennas.

Location	2.5 dBi Antenna	6dBi Antenna
Point A	RSSI: −85 dBm	RSSI: −80 dBm
	Data Rate: 100 kbps	Data Rate: 120 kbps
Point B	RSSI: −90 dBm	RSSI: −75 dBm
	Data Rate: 80 kbps	Data Rate: 150 kbps
Point C	RSSI: −95 dBm	RSSI: −70 dBm
	Data Rate: 60 kbps	Data Rate: 180 kbps

## Data Availability

Data are contained within the article.

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
