# Peer review of "Implementation of an Intelligent Trap for Effective Monitoring and Control of the Aedes aegypti Mosquito"

_sensors, 2024, doi:10.3390/s24216932_

Round 1
Reviewer 1 Report
Comments and Suggestions for Authors
See attached file.

Reviewer 2 Report
Comments and Suggestions for Authors
The authors focused on an interesting topic of mosquito monitoring and deployed YOLOv7 on an IoT system. Here are some issues:
1. "Introduction" and "related works" section lacks simplicity.
2. Why is the dataset composed of 3 categories of insects since the system is designed for mosquito monitoring. Will introducing negative samples improve the performance?
3. The authors merely deployed YOLOv7 on a Raspberry Pi based system. What is the innovation and originality of this system?
4. There are still some typos in the manuscript. Please check the format and spelling thoroughly.
Reviewer 3 Report
Comments and Suggestions for Authors
The idea presented is quite good but the scientific soudness is not so good, the whole work being very much like a students' project for the purpose of training, exercising rather than an academic paper. Some suggestions for improvement:
1) Figures should be drawn using standard software rather than pictures, e.g. Figs 3, 4 should be drawn with CAD, Visio with dimensions, with necessary accessories attached like camera;
2) In section 3.3.1, actual images taken by the onboard camera should be presented, from where the algorithm should be explained in detail as to how this dection is implemented, with flowchart and necessry figures showing the whole process. Section 3.3.2 should not have been simply an introduction of YOLOv7, rather a detailed flowchart used by this paper;
3)How this trap works? with the camera covering a lagre area(? how large, no idea since no installation is shown), and a small hole on the side of the trap, how to ensure the insects detected by the camera be flown in/out of this hole properly;
4)On line 440-443, the paper presents the experiment results without giving detailed steps or procedures. Not convincing.
Round 2
Reviewer 1 Report
Comments and Suggestions for Authors
The paper notably improved after its revisions, and I have no further comments.
Author Response
The paper notably improved after its revisions, and I have no further comments.
We thank the Reviewer for the comments and for the careful review of the manuscript. Your comments helped to create a better version of the article
Reviewer 2 Report
Comments and Suggestions for Authors
My concerns have been responsed. Now I have no questions on the scientiics of this manuscript. The introduction and conclusion can be further improved.
Comments on the Quality of English LanguageThe quality of english is ok for most readers.
Author Response
My concerns have been responsed. Now I have no questions on the scientiics of this manuscript. The introduction and conclusion can be further improved.
We thank the Reviewer for the comments and for the careful review of the manuscript. Your comments helped to create a better version of the article. We improved the introduction and conclusions sections in the final version.
Reviewer 3 Report
Comments and Suggestions for Authors
1)In line 235 the paper says "The trap detects the presence of bees through sensors", which sensors are they and where, how they are installed should be explained and shown in Figs 2, 5 or better 3 and 4 too;
2)Figure numbering in the text should be checked carefully. In line 176 " Figure 3"most probably be "Figure 2", and the "Figure 4.9"in line 215 must be wrong.
Author Response
We thank the Reviewer for the comments and for the careful review of the manuscript. Your comments helped to create a better version of the article.
1)In line 235 the paper says "The trap detects the presence of bees through sensors", which sensors are they and where, how they are installed should be explained and shown in Figs 2, 5 or better 3 and 4 too;
The authors thank the reviewer for this valuable feedback. We change the description in the article to:
The trap detects the presence of bees by using a camera attached to a Raspberry Pi 4 board, with the YOLOv7 algorithm running, which is responsible for identifying the bees, and the CPU activates the fans, generating an airflow that directs the insect towards the trap entrance.
2)Figure numbering in the text should be checked carefully. In line 176 " Figure 3"most probably be "Figure 2", and the "Figure 4.9"in line 215 must be wrong.
The authors thank the reviewer for this valuable feedback. We correct the references of figures.